# Nutritional Quality and Fatty Acids Composition of Invasive Chinese Mitten Crab from Odra Estuary (Baltic Basin)

**DOI:** 10.3390/foods12163088

**Published:** 2023-08-17

**Authors:** Przemysław Czerniejewski, Grzegorz Bienkiewicz, Grzegorz Tokarczyk

**Affiliations:** 1Department of Commodity Science, Quality Assessment, Process Engineering and Human Nutrition, West Pomeranian University of Technology, Papieża Pawła VI 3, 71-459 Szczecin, Poland; pczerniejewski@zut.edu.pl (P.C.); gbienkiewicz@zut.edu.pl (G.B.); 2Department of Fish, Plant and Gastronomy Technology, West Pomeranian University of Technology, Papieża Pawła VI 3, 71-459 Szczecin, Poland

**Keywords:** Chinese mitten crab, edible yield, nutrition value, source of fatty acids

## Abstract

The Chinese mitten crab (*Eriocheir sinensis*) is recognized as an invasive species in Europe but increasing fishing efforts may hold economic benefits and yield positive ecological and nutritional outcomes. The objective of this study was to determine the yield of edible parts and the compositional and nutritional characteristics of this crab, especially as a source of n-3 PUFA. The overall yield of edible parts amounted to 38.09%, with males (27.72%) exhibiting a higher meat content compared to females (25.30%). The gonads displayed the highest protein content (24.12%), while the hepatopancreas (11.67%) showcased the highest fat content. Furthermore, the fatty acid composition varied depending on the distribution within different crab segments and gender and individual size. Notably, the gonad lipids contained the most nutritionally valuable n-3 fatty acids, followed by muscle and hepatopancreas lipids. The determined index of atherogenicity (IA) from 0.2 for gonadal lipids to 0.42 for hepatopancreas lipids, index of thrombogenicity (IT) in the range of 0.10 for gonads to 0.41 for hepatopancreas, and flesh lipid quality (FLQ) from 6.9 for hepatopancreas to 23 for muscle lipids indicate their pro-health properties. The ratio of n-3 to n-6 fatty acids showed Chinese mitten crab as an excellent source of oil that can be used for food fortification and dietary supplement production.

## 1. Introduction

The Chinese mitten crab (*Eriocheir sinensis*) has a natural distribution range that stretches from the Fujian province in China (26° N) to the Korean Peninsula (40° N) [1,2]. In European waters, the first sightings of this species occurred during fishing activities in the River Aller, Germany, back in 1912. It is believed that young mitten crabs were inadvertently transported to new areas in the ballast tanks of ships, where they found ideal conditions for survival and reproduction [2]. Subsequently, they made their way to the Baltic Sea through the Kiel Canal in 1927 [3].

Throughout the 1930s and 1940s, their presence expanded beyond the North Sea drainage basin (Denmark, The Netherlands, Belgium, France, and the UK), reaching the Baltic Sea drainage basin (Poland, Sweden, and Finland) [2,4,5]. By the 1960s, they had even colonized the Mediterranean Sea drainage basin [2,6]. Their range further expanded to include Spain, Portugal, and eastern Europe by the late 19th century [2,7,8]. This invasive trend continued in the 1960s to North America [9], and more recently, in the early 21st century, they have been observed in Iran in western Asia [10]. 

The rapid spread of the Chinese mitten crab into new territories, coupled with the significant impact it has on native aquatic fauna habitats [8,11], has earned it a notorious reputation as “one of the world’s 100 worst invasive alien species” [12]. Consequently, EU regulations [13] prohibit the release of these crabs back into surface waters within member states. As a result, captured mitten crabs in European waters are usually discarded, despite their potential value in the food industry. 

Interestingly, in their native range, these crabs have long served as a traditional food source [11] and have been successfully cultivated in aquaculture for over three decades [14]. The controlled production of Chinese mitten crabs has experienced a remarkable increase, skyrocketing from 3305 tons in 1989 to a staggering 808,000 tons in 2021 [14,15]. This incredible growth has earned them the title of “the most important crab in the world,” representing nearly 70% of China’s crab aquaculture production [14,16]. 

Within their natural range, these crabs are renowned for their high protein content, well-balanced amino acids, and lipids rich in n-3 polyunsaturated fatty acids (PUFA), all contributing to their unique aroma and desirability among consumers [17,18]. Additionally, Chinese mitten crabs are prized for their high edible yield, with approximately 33% of their body being considered suitable for consumption [17], including the highly sought-after delicacy of their gonads [11]. Therefore, the most highly prized individuals are those caught during the reproductive migration period [9], although the meat and hepatopancreas also have excellent flavor and nutritional value [16].

In European waters, the annual catch of Chinese mitten crabs reaches approximately 180–200 tons, with around 4–5 tons being harvested specifically from the Odra Estuary [3,8]. There have been no specific investigations conducted on the lipid levels and fatty acid composition of the edible portions of European Chinese mitten crabs, apart from general data provided by Fladung [3]. However, it is very likely that, similar to their counterparts in their native habitats, the edible parts of these crabs could be effectively utilized. Expanding our knowledge in this regard becomes crucial, as highlighted by Wang et al. [19], as substantial variations exist in the flavor quality of *E. sinensis* sourced from different locations, influenced by environmental factors and the crabs’ diet. Moreover, according to Nędzarek and Czerniejewski [20], the meat of Chinese mitten crabs from European waters can serve as an excellent source of essential nutrients, with concentrations of non-essential and potentially toxic trace elements remaining well below the recommended dietary guidelines.

The primary objective of this study was twofold: first, to determine the yield of edible parts and compositional characteristics of Chinese mitten crabs through a comprehensive analysis of their fatty acid profile, and second, to assess the nutritional quality of the edible portions of wild Chinese mitten crabs originating from a non-native population found in the Odra Estuary within the Baltic basin.

## 2. Materials and Methods

### 2.1. Study Area and Sampling Procedure

The Chinese mitten crabs (*Eriocheir sinensis*) used for the research were captured during the period from 18 October to 28 October 2022 and comprised 27 females and 29 males, for a total of 56 individuals. During this period (autumn), Polish fishermen catch the most Chinese mitten crabs. The catches were obtained using fishing tools in the waters of the Oder Estuary, specifically spanning from Lake Dąbie to the mouth of the Piastowski Canal and leading to the Szczecin Lagoon (see Figure 1). Immediately following the collection, the crabs were transported in Styrofoam boxes with ice at a temperature of about 0 °C to the laboratory for further examination. The transport time was about 40 min.

Upon arrival at the laboratory, the captured crabs were carefully weighed using a precise electronic scale, ACA1000 (Axis, Gdańsk, Poland) with an accuracy of 0.01 g. To determine the sex of each crab, their abdominal structures were examined and classified according to the established method introduced by Czerniejewski [21]. Additionally, the carapace length (CL) and width (CW) of each crab were meticulously measured using an 1108-200-USB INSIZE 8″ electronic caliper (Gaging & Software Technologies, Inc., Colorado Springs, CO, USA) coupled to a computer system, ensuring an accuracy of ±0.01 mm.

Following these initial procedures, the meat samples were meticulously extracted from the claws, and the hepatopancreas and female gonads were also collected. All samples of edible parts during transport were preserved in ice (0 °C), and in the laboratory, the edible parts were stored at a temperature below −20 °C for further analysis. 

### 2.2. Body Condition

The condition of crabs (K) was determined using the Fulton coefficient [21], according to the equation:(1)$$K=\frac{W}{{CW}^{3}}\times 100$$
where W is the weight of the Chinese mitten crab (in g) and CW is the carapace width (in cm).

### 2.3. Analysis of General Composition and Edible Yield

The moisture content was determined by drying the sample in an oven at 105 °C until a constant weight was achieved [22]. The crude protein content was determined using the Kjeldahl method [22], and a conversion factor of 6.25 was applied to convert the total nitrogen to crude protein. The ash content was determined by ashing the samples in an oven at 550 °C for 8–12 h [22].

To calculate the gonadosomatic index (GSI), hepatosomatic index (HSI), meat yield (MY), and total edible yield (TEY), the ovaries, hepatopancreas, and meat from each crab were weighed. The formulas provided by Shao et al. [23] and Long et al. [24] were used for the calculations.

### 2.4. Fat Content and Fatty Acid Composition

The total lipid content was extracted using a chloroform/methanol (2:1, *v*/*v*) mixture following the method described by Folch et al. [25]. The fat content was determined through gravimetric analysis by evaporating the solvent from a specific volume of the lipid extract. The fat content was then calculated as grams per 100 g of the sample.

To obtain the fatty acid methyl esters (FAMEs), the extracted lipids underwent alkaline hydrolysis using 0.5 N sodium methylate (CH_3_ONa) [26]. Subsequently, the FAMEs were separated using a gas chromatography apparatus coupled with a mass spectrometer (Agilent Technologies 7890A, Santa Clara, CA, USA) and equipped with a split/splitless type injector. The separation conditions for the FAMEs were: SPTM 2560 column (100 m length, 0.25 mm inner diameter, and 0.20 μm film thickness, catalog no. 24056); helium was used as the carrier gas with a constant flow rate of 1.2 mL/min; the split ratio was 1:50; the injector and detector temperatures were set at 220 °C; and the programmed furnace temperature was started at 140 °C for 5 min, then ramped up to 240 °C at a rate of 4 °C/min. The total analysis time was 45 min.

The qualitative interpretation of chromatograms was based on comparing the retention times and mass spectra of the specific FAMEs in the sample with those of analogous FAME standards provided by Sigma Company, Kawasaki, Japan (Lipid Standard). C19:0 (Merck, Poland) was used as the internal standard for quantification purposes.

### 2.5. Index of Atherogenicity (IA)

The atherogenic potential of FAs was determined using the index of atherogenicity (IA) [27], according to the equation: (2)$$IA=\frac{C12:0+\left(4\times C14:0\right)+C16:0}{\Sigma UFA}$$

### 2.6. Index of Thrombogenicity (IT)

The thrombogenic potential of FAs was determined using the index of thrombogenicity (IT) [27], according to the equation:(3)$$IT=\frac{C14:0+C16:0+C18:0}{\left(0.5\times \Sigma MUFA\right)+\left(0.5\times \Sigma n-6PUFA\right)+\left(3\times \Sigma n-3PUFA\right)+(n-3/n-6)}$$

### 2.7. Polyunsaturated Fatty Acid/Saturated Fatty Acid Ratio (PUFA/SFA)

The impact of diet on cardiovascular health was determined using the ratio of polyunsaturated fatty acid/saturated fatty acid (PUFA/SFA) [28], according to the equation:(4)$$PUFA/SFA=\frac{\Sigma PUFA}{\Sigma SFA}$$

### 2.8. Fish Lipid Quality/Flesh Lipid Quality (FLQ)

The quality of lipids was determined using the index of lipid quality/flesh lipid quality (FLQ) [28], according to the equation:(5)$$FLQ=\frac{100\times (C22:6n-3+C20:5n-3)}{\Sigma FA}$$

### 2.9. Statistical Analysis

Differences in the mean values of the examined traits between males and females were analyzed using the Mann–Whitney U test. Additionally, differences among the meat, hepatopancreas, and gonads were assessed using a one-way analysis of variance (ANOVA). The post hoc Tukey test was employed for further comparisons. Statistical significance was considered at *p* < 0.05. All statistical analyses were conducted using Statistica 13.1 software [29].

## 3. Results

### 3.1. Size and Condition Coefficient of the Chinese Mitten Crab

The mean carapace width of the Chinese mitten crab was 67.91 ± 7.82 mm and the mean weight amounted to 148.93 ± 52.9 g. Among the 56 crabs, 18 crabs had a carapace width of 55–60 mm and 38 crabs had a width of 70–75 mm. In the carapace length class of 55–60 mm, 9 females and 9 males were tested, while in the 70–75 mm class, 18 females and 20 males were tested. Males exhibited a similar carapace width and slightly higher individual weight (see Table 1). Additionally, the mean value of Fulton’s condition factor (K) was significantly higher in males compared to females.

### 3.2. The Proximate Composition of CMC and Estimated TEY, GSI, HSI, and MY

Statistically significant differences were observed in the moisture, protein, fat, and ash content among the different edible parts of the Chinese mitten crab (see Table 2). The meat exhibited the highest moisture content (78.31%) while having the lowest fat (0.87%) and ash (1.41%) content. In contrast, the hepatopancreas contained the highest amount of fat (11.67%) compared to the other edible parts. The female gonads displayed the highest protein content (24.12%), whereas the hepatopancreas had the lowest protein content (13.62%).

The overall yield of edible parts of Chinese mitten crabs amounted to 38.09%, with the meat yield contributing to 26.63% and the hepatosomatic index (HSI) to 8.82%. No statistically significant differences were found between males and females in terms of the total edible yield (TEY) and HSI. However, females demonstrated higher values of the gonadosomatic index (GSI) and lower meat yield (MY) values compared to males (see Table 3).

### 3.3. Fat Content in Females and Males of CMC in Different Size Classes

Noteworthy differences were observed in the fat content among the various edible parts of both male and female crabs. The highest fat content was consistently found in the hepatopancreas, followed by the gonads and meat (see Table 4). Furthermore, females exhibited a statistically significant higher fat content in both the meat and hepatopancreas compared to males (*p* < 0.05). Additionally, when examining the crabs by size classes, a decreasing trend in fat content was observed with increasing size in all examined edible parts of females, but in the meat of males, an increasing trend in fat content was noted.

### 3.4. Fatty Acid Composition

A total of 18 fatty acids were identified and quantified in the extracted hepatopancreas fat. The overall content of fatty acids ranged from 715 mg FA/gram of fat in the larger male group (70–75) to 859 mg FA/gram of fat in the larger female group, showing significant differences between the size groups of females and also between the sexes within both size groups (see Table 5). The quantity of n-3 polyunsaturated fatty acids (PUFA) was found to be size-dependent, with larger individuals exhibiting significantly higher levels of these fatty acids in their hepatopancreas. This relationship was particularly prominent for nutritionally valuable fatty acids such as EPA and DHA, although it was more pronounced for EPA, which demonstrated higher levels compared to DHA.

In the case of the n-6 PUFA group of fatty acids, significant differences were observed depending on the sex, and in males, also depending on the size. The total amount of these fatty acids ranged from 141.65 mg FA/g of fat in large males to 192.91 mg/g of fat in small males (see Table 5). This relationship translated into the ratio of n-3 to n-6 fatty acids. On average, this ratio was 0.6 in females and ranged from 0.35 in small individuals to 0.59 in large individuals in males. The FAO/WHO recommends a dietary ratio of n-3 to n-6 PUFA of at least 0.2–0.25, with a higher ratio (>0.25) being more favorable in human nutrition. This means that the hepatopancreas of crabs with the largest sizes has a higher nutritional value. 

In the meat of the crabs, the total sum of fatty acids ranged from 345 to 472 mg/g of fat, which was nearly half the amount compared to the hepatopancreas lipids (see Table 5 and Table 6). The total fatty acids differed significantly depending on the sex but no significant differences were found based on size groups. The total amount of n-3 PUFA differed significantly between sexes within the same size group and in males between each carapace class. The content of this group of fatty acids in females and males was under 100 mg FA/g of fat. FLQ is more suitable for marine products given their higher proportions of EPA and DHA. When evaluating it in hepatopancreas lipids, it was found that its level in females was balanced in the analyzed size classes and amounted to less than 8.5. However, clear differences were observed between size classes in males. Small males were characterized by an FLQ index at the level of 6.92 and the large ones at 9.01. This also translated into the amount of EPA and DHA acids per 100 g of raw material, which is where these differences were even more visible. Lower values of indices of IA and IT were found in males compared to females (see Table 5).

In the case of muscle lipids, similar to hepatopancreas lipids, there were notable differences in the content of EPA and DHA. Significant variations in the levels of these fatty acids were observed between the sexes and size groups of the studied crabs (see Table 6). The n-6 PUFA group of fatty acids also exhibited significant differences depending on the size, while no significant differences were found between the sexes.

Assessing the nutritional value of muscle fat in crabs, particularly the ratio of n-3 to n-6 fatty acids, it was found to be more favorable in the examined edible portion compared to the hepatopancreas lipids. This was especially evident in the muscle lipids of larger individuals, with a ratio of 1.83 in females and 1.73 in males. However, considering the fat content in the muscles and hepatopancreas (see Table 6), it is challenging to consider the muscle of the Chinese mitten crab as a significant source of n-3 PUFA in the diet; it provided over ten times less EPA and DHA. When accounting for the yield of pure meat, it becomes apparent that the muscle of the Chinese mitten crab may not be a sufficient source of n-3 PUFA. The determined IA and IT indices for the muscle lipids were lower by more than 20% compared to hepatopancreas lipids, both in females and males. On the other hand, the FLQ indices were more than twice as high. Also, the amount of EPA and DHA acids per 100 g of the product differed significantly between muscle lipids and hepatopancreas lipids. Comparing the indices between size classes and sex, it was found that, as in the case of hepatopancreas lipids, the least favorable indices for FLQ and the content of EPA and DHA acids also apply to small males (see Table 5 and Table 6). 

The analysis of fatty acid composition in the gonads revealed significant differences in the total fatty acid content (see Table 7). The gonads of larger females contained a higher number of fatty acids (590 mg per gram of fat), while smaller females had less fatty acids (568 mg per gram of fat). Statistically significant differences concerned the group of SFA and MUFA acids, the higher content of which was found in the gonads of large females. Conversely, there were no significant differences in the levels of the highly nutritionally valuable EPA and DHA fatty acids based on the size of individuals. Both smaller and larger females exhibited approximately twice as much EPA as DHA. It is worth noting the favorable ratio of n-3 to n-6 fatty acids, averaging 1.52 in the gonads, compared to a lower average ratio of 0.57 in the hepatopancreas (see Table 2) across the size groups of the crabs tested. There were no differences in the IA and IT indices between the lipids from the gonads of small and large females; however, lipids from the gonads of small females were more favorable in terms of nutrition. The FLQ index and the amount of EPA and DHA per 100 g of the product concerning lipids from the gonads of small females were more favorable than those found for lipids from the large female gonads (see Table 7). 

When analyzing the content of SFA, MUFA, and PUFA in the edible parts of the crab, statistically significant differences were found in both female and male individuals (see Table 8). In the group of females, the highest amounts of SFA, MUFA, and PUFA were found in the hepatopancreas. The least amount of SFA was found in female gonads, while MUFA and PUFA were found in meat. In the group of males, most SFA, MUFA, and PUFA were also found in the hepatopancreas. In comparing the content of these acids between males and females, it was found that, statistically significantly, more SFA and MUFA acids were found in the hepatopancreas and meat of females than in males. In the case of PUFA, their content in the hepatopancreas and meat of females was higher than in males but was statistically insignificant. The highest content of n-3 PUFA in the group of females was found in the gonadal lipids and the lowest in the hepatopancreas, while the difference in their content in relation to the meat was statistically insignificant. In the group of males, the amount of n-3 PUFA was higher in the meat than in the hepatopancreas, but it was also statistically insignificant. There were no statistically significant differences in the content of n-3 PUFA between females and males, although their content was higher in the group of females. Statistically significant differences in the content of n-6 PUFA in the analyzed edible parts of crabs were found among both females and males. The most acids of this group in females and males were found in the hepatopancreas, with significantly more of them in males. The least n-6 PUFAs were found in the meat of males and females, but these differences were not statistically significant. The n-3/n-6 ratio varied depending on the edible parts of the crabs analyzed, with the highest values noted for hepatopancreas (2.25 for males and 1.75 for females). For other parts of the crabs tested, the value of this coefficient was much lower. In the case of female gonads, the value of the n-3/n-6 coefficient was 0.66, while for the meat, a significant difference in the value of this coefficient was observed between females and males—0.57 and 0.68, respectively. 

## 4. Discussion

Due to their protein and lipid content, aquatic food sources are an important source of nutrition that meets human dietary needs [30]. In recent years, the high catch of aquatics and increased demand for aquatic foods have led to the intensive development of aquaculture and attempts to utilize less economically valuable raw materials and invasive species. An example is the use of Chinese mitten crab for nutritional purposes. The carapace width of the crabs captured for this study (females 67.21 ± 6.98 mm, males 68.57 ± 7.95 mm) was consistent with the size range of this species previously caught in the Odra River (Poland): 45–90 mm [8,21] and the Tagus River (Portugal): 60–90 mm [7].

Jiang et al. [31] reported that the edible portion of the Chinese mitten crab accounts for 39.50–44.9% of the crab’s total weight, while Chen et al. [17] reported a content of 33.4%. In the case of the examined crabs from the Odra River estuary, the proportion of edible parts was found to be 36.8–37.7%, primarily composed of meat (25.2–27.7%), hepatopancreas (8.3–9.1%), and gonads (0.19–5.2%). When compared to Chinese aquaculture populations, crabs from the Odra River estuary exhibited a higher proportion of hepatopancreas but smaller gonads in females [17,31]. These differences may be attributed to variations in the gonad maturity stage, diet composition [23], and water environmental conditions [32]. It may also be due to the fact that they are not fully ready for reproduction because, as observed by Long et al. [24] and Jiang et al. [31], the increase in GSI is accompanied by a decrease in HSI, indicating the transfer of lipids from the hepatopancreas to developing gonads [33]. 

From a technological resource efficiency perspective, the higher meat yield (MY) observed in males compared to females is significant. This disparity is a result of the sexual dimorphism in Chinese mitten crabs, which is manifested not only in the shape of the abdomen but also in the larger claws found in males [34], leading to a greater quantity of meat.

When evaluating the nutritional quality of the edible parts of the Chinese mitten crab, the biochemical composition [31,33] plays a crucial role, as it is influenced by tissue type and undergoes seasonal changes associated with physiological shifts [19]. Our research has revealed that the muscles are a structural tissue characterized by a high protein content (19.41%) and low fat content (0.87%), which aligns with previous findings reported for this species by Chen et al. [17] and Wang et al. [19]. The hepatopancreas, a multifunctional organ responsible for lipid and cholesterol absorption and storage [35,36], exhibits a high lipid content ranging from 15.01% to 45.09% in Chinese mitten crabs [19,24], while the protein content is relatively low, ranging from 6.29% to 10.28% [19,31]. In the examined crabs from the Odra River estuary, the fat content in the hepatopancreas was lower (11.67%), while the protein content was higher (13.62%). These slight differences can be attributed to physiological changes during the maturation process [24] or variations in environmental conditions and diet [36].

Typically, the female gonads are the better source of protein and amino acids compared to the hepatopancreas [31] and contain 18.69% to 29.74% protein [19] and 8.72% to 16.20% fat [24,31]. However, in the examined crabs from the Odra River estuary, the fat content in the gonads was lower (6.05%), while the protein content was within the expected range (24.12%). The lower fat content in the gonads and the low gonadosomatic index of these female crabs (5.20%) indicated that the examined individuals were not fully mature. Previous studies by Wen et al. [37] and Wu et al. [33] suggested that when the ovaries of female *E. sinensis* reach full maturity and become hardened, the lipid levels in the gonads significantly increase.

The fatty acid composition of the edible parts in aquatic organisms serves as a crucial indicator for assessing their nutritional quality [38,39]. In the case of Chinese mitten crabs, studies have been conducted on both aquaculture-raised specimens and those found in their natural habitats [24,40]. However, there is a lack of information regarding the quantity and composition of fatty acids in individuals inhabiting areas beyond their natural distribution range, characterized by distinct environmental conditions and food resources.

Analyzing the fatty acid content in Chinese mitten crabs from the Odra River estuary revealed the presence of 18 fatty acids, with a total amount ranging from 345.85 to 715.58 mg FA/gram of fat depending on the specific edible part of the crab. Similarly, investigations conducted by Celik et al. [41] on the Blue crab (*Callinectes sapidus*) from the northeast Mediterranean, as well as by Ying et al. [42] on Chinese mitten crabs from Chinese waters, demonstrated significant variations in the content and composition of fatty acids among the different edible parts of these organisms.

In our study, we observed the lowest levels of fatty acids in the muscle lipids, which were approximately half the amount found in the hepatopancreas lipids. This discrepancy can be attributed to the selective absorption of dietary lipids by the hepatopancreas before their transportation to the muscles [43]. Furthermore, muscle lipids in crabs, similar to lean fish meat, exhibit higher levels of phospholipids and non-saponifiable sterols [38]. In contrast, hepatopancreas lipids primarily consist of acylglycerols serving as an energy source. This assertion is further supported by the varying amounts of EPA and DHA, with higher quantities present in muscle lipids, particularly in the phospholipids that constitute cellular membranes. Despite the low fat content, muscle lipids were characterized by lower IA and IT indices compared to hepatopancreas lipids and slightly higher indices than gonadal lipids. The atherogenic index (IA) and thrombogenicity index (IT) are lipid quality indicators, and they are determined based on the relative contents of particular groups of fatty acids. These indices indicate the overall dietary quality of lipids and their potential effects on the development of coronary disease [38]. The IA and IT indices values were lower than those obtained for other types of food, such as beef, chicken, lamb, pork, and rabbit [28], but higher than in lobster muscle (IA ranged from 0.22 to 0.26 and IT ranged from 0.14 to 0.17) [44] and crab meat (IA = 0.17 and IT = 0.12) [45]. On the other side, the IA indices of 0.27 to 0.34 obtained in our work were comparable to the indices of shellfish lipids. In the lipids of scallops, *Chlamys farreri*, this coefficient ranged from 0.31 to 0.37, and in the lipids of *Patinopecten yessoensis*, from 0.29 to 0.35 [46]. However, the IT index in the range of 0.22–0.29 was higher compared to the IT index for shellfish lipids (from 0.09 to 0.17) [46], but similar to the lipids of shrimps [47] or fish [48]. 

Moreover, our analysis of Chinese mitten crabs from the Odra River estuary indicated that the fatty acid content was influenced by the sex and size of the individuals. Notably, the hepatopancreas of females exhibited the highest amounts (over 800 mg FA/gram of fat), while males recorded approximately 700 mg FA/gram of fat. These variations can be attributed to the physiological state of the organism and metabolic processes occurring primarily in the gonads, muscles, and hepatopancreas, which are influenced by factors such as maturity stage and diet [19,24].

The fatty acid composition in crabs is influenced by the type of fatty acids present in their diet [40], which in turn affects their growth, reproduction, and nutritional value [49]. Chen et al. [17] reported that monounsaturated fatty acids (MUFA) dominated the fatty acid profile of Chinese mitten crabs, accounting for 49.8% of the total fatty acids. However, in crabs from the Odra River estuary, the percentage of MUFA, although dominant, was slightly lower and varied depending on the edible part, ranging from 36.4% in the muscle of large males to 46% in the gonads of large females. In Chinese populations of Chinese mitten crabs, saturated fatty acids (SFAs) and polyunsaturated fatty acids (PUFAs) account for 24.9% and 23.9%, respectively [17]. Furthermore, He et al. [36] reported that the SFA content in the lyophilized edible parts of Chinese mitten crabs was 16.6% in females and 24.53% in males, with palmitic acid (C16:0) being the dominant fatty acid. Our study on crabs from the Odra River estuary indicated a slightly higher percentage of SFAs in all edible parts (averaging 27.51%). Meanwhile, the average percentage of PUFAs in the crabs we examined (34.04%) was similar to the data from Chinese waters. For example, He et al. [36] reported that the PUFA content ranged from 26.34% in males to 34.74% in females. 

Furthermore, our research demonstrated that the overall amount of SFAs, MUFAs, and PUFAs depends on their distribution in different crab parts and varies according to the sex and size of the crab. The highest content of MUFAs and PUFAs was typically found in the hepatopancreas, gonads, and muscles, and it tended to be higher in larger individuals. Additionally, the content of these fatty acids was generally higher in females than in males. The differences in the fatty acid content among individuals of different sizes may be attributed not only to size but also to the maturity stage of the gonads and the physiological state [50]. For example, Dvoretsky et al. [51] reported that the size of female red king crabs (*Paralithodes camtschaticus*) did not affect the fatty acid content in their edible parts, which is consistent with the findings of Ying et al. [42] on Chinese mitten crabs. When evaluating the ratio of PUFA to SFA fatty acids, it was found that the most favorable ratio in nutritional terms characterized gonadal lipids: 1.74 and 1.64 (small and large females, respectively), while the least favorable ratio was for hepatopancreas lipids, which, depending on sex and size, was between 0.91 and 1.19. PUFA/SFA is the most commonly used index for evaluating the nutritional value of dietary foods such as seaweed (0.42–2.12, except for *Gracilaria changii*), meat (0.11–2.042), fish (0.50–1.62), and shellfish (0.20–2.10) [28].

European diets tend to have lower levels of n-3 fatty acids compared to n-6 fatty acids [52]. This indicates the importance of consuming foods with a high n-3/n-6 ratio for optimal nutrition. The n-3/n-6 ratio serves as a valuable indicator for comparing the relative nutritional value of fats from different food sources, and a higher n-3/n-6 PUFA ratio is often associated with superior nutritional value. Experts from the FAO and WHO recommend a daily intake of at least 500 mg of n-3 fatty acids to achieve an n-3/n-6 PUFA ratio of approximately 1:5 [53]. Taking into account the FAO/WHO recommended daily intake of EPA and DHA acids in the range of 0.250–2 g, the amount provided by crab muscles is about 10 times lower. This is also evidenced by the ratio of the sum of mg EPA and DHA per 100 g of the product (see Table 8). Wang et al. [19] emphasized the significance of these parameters in their study on wild mitten crabs from three river basins.

In the examined samples of Chinese mitten crabs, the n-3/n-6 PUFA ratio averaged 0.53 for hepatopancreas lipids and ranged from 1.27 for small males to 1.83 for large females for muscle lipids. In the case of gonads, the ratio ranged from 1.46 to 1.60. These values are considerably lower compared to various species of marine crabs. For example, according to Dvoretsky et al. [51], the n-3 to n-6 PUFA ratio in the gonads of red king crabs from the Barents Sea exceeds 4. When comparing these values to Chinese aquaculture crabs, the ratio ranged from 0.5 in crabs fed with frozen waste to 1.1 in those fed with formulated feed [54]. In Chinese natural waters, the ratio varied from 0.54 to 1.7 depending on the maturity stage of the gonads [42]. It is worth noting that the n-3 PUFA content and the n-3/n-6 value in Chinese mitten crabs are lower compared to other crab species [51].

Despite the relatively high content of the nutritionally valuable EPA and DHA fatty acids in the edible parts of Chinese mitten crabs from the Odra River estuary, they cannot be considered a significant, high source of these acids in the human diet. With the EPA and DHA levels in muscles averaging 80 mg/g of fat, and considering a fat content of 1%, one would need to consume over 500 g of crab meat to meet the recommended daily intake of these fatty acids. The determined index of the sum of mg of EPA and DHA acids per 100 g of the product indicated that the best source of them were hepatopancreas lipids (over 850 mg/100 g), with the exception of small males (423 mg/100 g), followed by gonads (average about 500 mg/100 g). However, for muscle lipids, this index was only from 50 mg/100 g for small males to 91 mg/100 g for small females. Such a wide range of results is consistent with the studies of Rincón-Cervera et al. [55] who evaluated the quantification and distribution of omega-3 fatty acids in South Pacific fish and shellfish species. Nevertheless, the favorable n-3 to n-6 ratio in the edible parts of Chinese mitten crabs, particularly in the gonads with a fat content of 6.05%, makes them suitable not only as a source of dietary energy but also for the production of supplements and dietary ingredients. 

## 5. Conclusions

The European mitten crab, although abundant in many populations, is not extensively utilized for culinary purposes in Europe, despite being considered a delicacy in its natural habitat. Our research indicated that this limited utilization may be attributed to the relatively low yield of edible parts, which accounts for only 38.09%, with a meat yield of 26.63%. Consequently, it struggles to compete as a viable food source compared to other aquatic resources. However, the remarkable fatty acid composition and the presence of valuable nutritionally significant n-3 fatty acids, particularly EPA and DHA, which are long-chain LC-PUFAs, along with the favorable n-3/n-6 fatty acid ratio found in the gonads and hepatopancreas, make this resource suitable for supplementary purposes. The nutritional indices calculated in this work, IA and IT, are the most commonly used to assess the composition of fatty acids because they provide important information on health and nutritional values, while EPA + DHA is commonly used to assess the nutritional quality of marine animal products. The determined IA indices from 0.2 for gonadal lipids to 0.42 for hepatopancreas lipids, IT in the range of 0.10 for gonads to 0.41 for hepatopancreas, and FLQ from 6.9 for hepatopancreas to 23 for muscle lipids indicate their pro-health properties. The ratio of n-3 to n-6 fatty acids showed Chinese mitten crabs to be an excellent source of oil that can be used for food fortification and dietary supplement production; they can be utilized in the production of dietary supplements and pharmaceutical preparations. Expanding their use in the food and pharmaceutical industries will lead to the increased harvesting of this invasive species in European waters, which is not only economically viable but also ecologically necessary to mitigate their impact on the aquatic environment. Additionally, the local sourcing of this resource ensures high freshness within the supply chain, which is crucial due to the rapid oxidation and loss of its valuable components, particularly the long-chain polyunsaturated fatty acids (LC n-3 PUFA).

## Figures and Tables

**Figure 1 foods-12-03088-f001:**
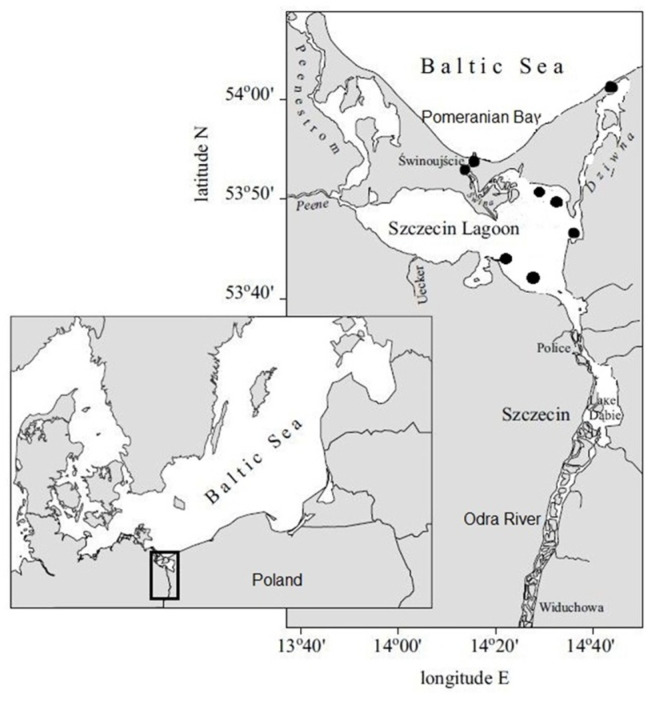
Map with markings indicating the locations of crab fishing.

**Table 1 foods-12-03088-t001:** The mean carapace width, body weight, and Fulton coefficient of the tested Chinese mitten crab; significance level of differences (*p*) determined by Mann–Whitney U test.

Sex	Number	Carapace Width (mm)	Weight (g)	Fulton Coefficient (K)
Female	27	67.21 ± 6.98	146.49 ± 43.65	0.43 ^b^ ± 0.06
Male	29	68.57 ± 7.95	151.21 ± 53.61	0.47 ^a^ ± 0.09

^a,b^—Significant differences between females and males (Mann–Whitney U test, *p* < 0.05). Weight and carapace width do not show any significant difference between the sexes.

**Table 2 foods-12-03088-t002:** Proximate composition (%) of the Chinese mitten crab.

	Moisture (%)	Protein (%)	Fat (%)	Ash (%)
Meat	78.31 ^a^ ± 0.84	19.41 ^a^ ± 0.53	0.87 ^a^ ± 0.10	1.41 ^a^ ± 0.02
Hepatopancreas	72.56 ^b^ ± 1.53	13.62 ^b^ ± 0.42	11.67 ^b^ ± 0.43	2.15 ^b^ ± 0.06
Gonads	68.21 ^c^ ± 1.23	24.12 ^c^ ± 0.46	6.05 ^c^ ± 0.34	1.62 ^a^ ± 0.04

^a–c^—Significant differences between the individual edible parts of the Chinese mitten crab (ANOVA, Tukey’s test, *p* < 0.05).

**Table 3 foods-12-03088-t003:** Total edible yield, hepatosomatic index, gonadosomatic index, and meat yield of adult *E. sinensis*.

Parameter (%)	Female	Male
Total edible yield (TEY)	38.70 ^a^ ± 2.46	37.70 ^a^ ± 2.62
Meat yield (MY)	25.30 ^a^ ± 1.65	27.72 ^b^ ± 1.92
Hepatosomatic index (HSI)	8.20 ^a^ ± 0.95	9.10 ^a^ ± 0.85
Gonadosomatic index (GSI)	5.20 ^a^ ± 0.62	0.90 ^b^ ± 0.11

^a,b^—Significant differences between females and males (Mann–Whitney U test, *p* < 0.05).

**Table 4 foods-12-03088-t004:** Fat content (%) in different parts of the Chinese mitten crab categorized by sex and carapace width (mm).

Parts of Crab		Female			Male	
	Whole Group	55–60 mm	70–75 mm	Whole Group	55–60 mm	70–75 mm
Gonads	6.05 ± 0.34	6.59 ^a^ ± 0.38	5.53 ^b^ ± 0.13	-	-	-
Meat	0.92 ^A^ ± 0.11	1.02 ^a^ ± 0.12	0.80 ^a^ ± 0.08	0.82 ^B^ ± 0.08	0.79 ^a^ ± 0.09	0.86 ^a^ ± 0.07
Hepatopancreas	12.63 ^A^ ± 0.62	13.14 ^a^ ± 0.79	12.24 ^b^ ± 0.27	10.71 ^B^ ± 0.52	8.34 ^a^ ± 0.60	13.17 ^b^ ± 0.27

^a,b^—Significant differences within one sex (Mann–Whitney U test, *p* < 0.05). A, B—Significant differences between females and males (Mann–Whitney U test, *p* < 0.05).

**Table 5 foods-12-03088-t005:** The content of individual fatty acids (expressed in mg FA/gram of fat) in the hepatopancreas according to the sex and carapace width (mm) of the individuals.

Fatty Acid	Female		Male	
	55–60 mm	70–75 mm	55–60 mm	70–75 mm
C12:0	0.32 ^aA^ ± 0.01	0.29 ^aA^ ± 0.01	0.30 ^aA^ ± 0.02	0.30 ^aA^ ± 0.01
C14:0	19.19 ^aA^ ± 0.31	17.67 ^bA^ ± 0.31	11.65 ^aB^ ± 0.18	15.77 ^bB^ ± 0.17
C15:0	7.04 ^aA^ ± 0.11	8.15 ^bA^ ±0.14	6.50 ^aB^ ± 0.10	7.29 ^bB^ ± 0.08
C16:0	156.73 ^aA^ ± 2.54	170.33 ^bA^ ± 3.02	136.33 ^aB^ ± 2.09	125.27 ^bB^ ± 1.33
C17:0	26.12 ^aA^ ± 0.42	36.41 ^bA^ ± 0.65	36.83 ^aB^ ± 0.56	41.86 ^bB^ ± 0.44
C16:1	129.37 ^aA^ ± 2.09	158.73 ^bA^ ± 2.82	76.57 ^aB^ ± 1.17	117.62 ^bB^ ± 1.25
C18:0	30.39 ^aA^ ± 0.49	30.35 ^aA^ ± 0.54	27.24 ^aB^ ± 0.42	27.17 ^aB^ ± 0.29
C18:1 n-9	188.79 ^aA^ ± 3.05	188.58 ^aA^ ± 3.34	169.87 ^aB^ ± 2.60	38.33 ^B^ ± 0.02
C18:2 n-6	47.21 ^aA^ ± 0.76	30.13 ^bA^ ± 0.53	74.89 ^aB^ ± 1.15	30.22 ^bA^ ± 0.32
C18:3 n-6	51.64 ^aA^ ± 0.84	60.66 ^bA^ ± 1.08	61.97 ^aB^ ± 0.95	56.82 ^bB^ ± 0.60
C20:1 n-9	9.86 ^aA^ ± 0.16	9.40 ^aA^ ± 0.17	7.88 ^aB^ ± 0.12	8.33 ^bB^ ± 0.09
C20:2 n-6	10.94 ^aA^ ± 0.18	8.79 ^bA^ ± 0.16	9.82 ^aB^ ± 0.15	8.0 5 ^bA^ ± 0.09
C20:4 n-6	25.03 ^aA^ ± 0.40	30.89 ^bA^ ± 0.55	28.94 ^aB^ ± 0.44	30.52 ^bA^ ± 0.32
C20:5 n-3	39.68 ^aA^ ± 0.64	44.00 ^bA^ ± 0.78	27.97 ^aB^ ± 0.43	37.20 ^bB^ ± 0.40
C22:4 n-6	4.71 ^aA^ ± 0.08	5.78 ^bA^ ± 0.10	6.46 ^aB^ ± 0.10	6.09 ^aB^ ± 0.06
C22:5 n-6	9.37 ^aA^ ± 0.15	11.24 ^bA^ ± 0.20	10.83 ^aB^ ± 0.17	9.96 ^bB^ ± 0.11
C22:5 n-3	16.29 ^aA^ ± 0.26	19.32 ^bA^ ± 0.34	17.39 ^aB^ ± 0.27	20.12 ^aA^ ± 0.21
C22:6 n-3	27.03 ^aA^ ± 0.44	28.81 ^aA^ ± 0.51	22.79 ^aB^ ± 0.35	27.27 ^bA^ ± 0.29
Total fatty acids	799.71 ^aA^ ± 12.94	859.52 ^bA^ ± 15.23	734.23 ^aB^ ± 11.24	715.88 ^aB^ ± 7.61
Σ SFA	239.80 ^aA^ ± 4.32	263.19 ^bA^ ± 5.22	218.85 ^aB^ ± 3.75	217.65 ^aB^ ± 2.59
Σ MUFA	328.02 ^aA^ ± 5.93	356.72 ^bA^ ± 7.07	254.33 ^aB^ ± 4.35	271.99 ^bB^ ± 3.23
Σ PUFA	231.89 ^aA^ ± 4.19	239.6 1 ^aA^ ± 4.75	261.05 ^aB^ ± 4.47	226.24 ^bB^ ± 2.69
Σ n-3 PUFA	83.00 ^aA^ ± 1.50	92.12 ^bB^ ± 1.83	68.14 ^aA^ ± 1.17	84.59 ^bB^ ± 1.00
Σ n-6 PUFA	148.89 ^aA^ ± 2.69	147.49 ^aA^ ± 2.92	192.91 ^aB^ ± 3.30	141.65 ^bA^ ± 1.68
n-3/n-6	0.56	0.63	0.35	0.60
DHA/EPA	0.68	0.65	0.81	0.73
IA	0.42	0.40	0.36	0.38
IT	0.42	0.41	0.41	0.36
PUFA/SFA	0.97	0.91	1.19	1.04
FLQ	8.34	8.47	6.92	9.01
mg EPA + DHA/100 g	876.46 ^aA^ ± 1.10	891.18 ^bA^ ± 1.32	423.28 ^aB^ ± 0.79	849.02 ^bB^ ± 0.70

^a,b^—Significant differences within each sex group based on size. ^A,B^—Significant differences observed between sexes within the same size categories.

**Table 6 foods-12-03088-t006:** The content of individual fatty acids in the meat {mg FA/gram of fat} depends on the sex and carapace width (mm) of individuals.

Fatty Acid	Female		Male	
	55–60 mm	70–75 mm	55–60 mm	70–75 mm
C12:0	0.27 ^aA^ ± 0.01	0.29 ^aA^ ± 0.02	0.025 ^aA^ ± 0.01	0.28 ^aA^ ± 0.03
C14:0	6.11 ^aA^ ± 0.15	6.2 2 ^aA^ ± 0.06	3.93 ^aB^ ± 0.06	3.69 ^aB^ ± 0.09
C15:0	2.92 ^aA^ ± 0.07	2.82 ^aA^ ± 0.03	2.70 ^aB^ ± 0.04	2.08 ^bB^ ± 0.05
C16:0	87.99 ^aA^ ± 2.18	79.99 ^bA^ ± 0.80	58.83 ^aB^ ± 0.83	56.45 ^aB^ ± 1.36
C17:0	15.48 ^aA^ ± 0.38	14.04 ^bA^ ± 0.14	17.22 ^aB^ ± 0.24	13.87 ^bA^ ± 0.34
C16:1	52.30 ^aA^ ± 1.30	46.09 ^bA^ ± 0.46	30.28 ^aB^ ± 0.43	34.72 ^bB^ ± 0.84
C18:0	27.47 ^aA^ ± 0.68	23.32 ^bA^ ± 0.23	17.32 ^aB^ ± 0.25	19.77 ^bB^ ± 0.48
C18:1 n-9	110.94 ^aA^ ± 2.75	102.08 ^bA^ ± 1.02	68.84 ^aB^ ± 0.97	79.36 ^bB^ ± 1.92
C18:2 n-6	16.78 ^aA^ ± 0.42	13.98 ^bA^ ± 0.14	16.81 ^aA^ ± 0.24	10.11 ^bB^ ± 0.24
C18:3 n-6	8.59 ^aA^ ± 0.21	7.64 ^bA^ ± 0.08	8.82 ^aA^ ± 0.12	5.61 ^bB^ ± 0.14
C20:1 n-9	9.62 ^aA^ ± 0.24	7.55 ^bA^ ± 0.08	5.89 ^aB^ ± 0.08	6.74 ^bB^ ± 0.16
C20:2 n-6	4.95 ^aA^ ± 0.12	3.76 ^bA^ ± 0.04	3.41 ^aB^ ± 0.05	2.60 ^bB^ ± 0.06
C20:4 n-6	23.95 ^aA^ ± 0.59	23.16 ^aA^ ± 0.23	26.92 ^aB^ ± 0.38	30.01 ^bB^ ± 0.73
C20:5 n-3	57.19 ^aA^ ± 1.42	53.93 ^bA^ ± 0.54	43.18 ^aB^ ± 0.61	55.45 ^bA^ ± 1.34
C22:4 n-6	1.36 ^aA^ ± 0.03	1.76 ^bA^ ± 0.02	1.69 ^aB^ ± 0.02	1.55 ^bB^ ± 0.04
C22:5 n-6	4.39 ^aA^ ± 0.11	2.84 ^bA^ ± 0.03	4.23 ^aA^ ± 0.06	2.61 ^bB^ ± 0.06
C22:5 n-3	10.12 ^aA^ ± 0.25	7.97 ^bA^ ± 0.08	8.66 ^aB^ ± 0.12	7.96 ^bA^ ± 0.19
C22:6 n-3	32.64 ^aA^ ± 0.81	35.10 ^bA^ ± 0.35	27.13 ^aB^ ± 0.38	27.22 ^aB^ ± 0.66
Total fatty acids	473.06 ^aA^ ± 13.13	432.53 ^aA^ ± 4.32	346.10 ^aB^ ± 4.88	360.10 ^aB^ ± 8.68
Σ SFA	140.23 ^aA^ ± 3.89	126.67 ^bA^ ± 1.42	100.26 ^aB^ ± 1.57	96.14 ^bB^ ± 2.60
Σ MUFA	172.85 ^aA^ ± 4.80	155.72 ^bA^ ± 1.74	105.01 ^aB^ ± 1.66	120.83 ^bB^ ± 3.26
Σ PUFA	159.97 ^aA^ ± 4.44	150.14 ^bA^ ± 1.68	140.84 ^aB^ ± 2.23	143.13 ^aA^ ± 3.87
Σ n-3 PUFA	99.96 ^aA^ ± 2.77	97.00 ^aA^ ± 1.09	78.96 ^aB^ ± 1.25	90.63 ^bB^ ± 2.45
Σ n-6 PUFA	60.01 ^aA^ ± 1.67	53.13 ^bA^± 0.59	61.88 ^aA^ ± 0.98	52.50 ^bA^ ± 1.42
n-3/n-6	1.60	1.83	1.27	1.73
DHA/EPA	0.57	0.65	0.63	0.49
IA	0.34	0.34	0.30	0.27
IT	0.29	0.28	0.25	0.22
PUFA/SFA	1.14	1.19	1.41	1.49
FLQ	19.00	20.60	20.33	22.97
mg EPA + DHA/100 g	91.63 ^aA^ ± 2.27	71.23 ^bA^ ± 0.91	55.54 ^aB^ ± 1.01	71.09 ^bA^ ± 2.04

^a,b^—Significant differences within each sex group based on size. ^A,B^—Significant differences observed between sexes within the same size categories.

**Table 7 foods-12-03088-t007:** The content of individual fatty acids {mg FA/gram of fat} in the gonads of females according to the carapace width (mm) of individuals.

Fatty Acid	Female	
	55–60 mm	70–75 mm
C12:0	0.26 ^a^ ± 0.01	0.25 ^a^ ± 0.03
C14:0	5.04 ^a^ ± 0.15	4.45 ^b^ ± 0.24
C15:0	2.92 ^a^ ± 0.09	2.92 ^a^ ± 0.16
C16:0	75.05 ^a^ ± 0.15	77.41 ^a^ ± 4.11
C17:0	17.83 ^a^ ± 0.54	19.57 ^b^ ± 1.04
C16:1	96.51 ^a^ ± 2.94	124.40 ^b^ ± 6.61
C18:0	17.46 ^a^ ± 0.53	16.92 ^a^ ± 0.90
C18:1 n-9	134.08 ^a^ ± 4.08	133.69 ^a^ ± 7.10
C18:2 n-6	21.41 ^a^ ± 0.65	14.76 ^b^ ± 0.78
C18:3 n-6	12.92 ^a^ ± 0.39	13.09 ^a^ ± 0.70
C20:1 n-9	13.50 ^a^ ± 0.41	11.41 ^b^ ± 0.61
C20:2 n-6	4.88 ^a^ ± 0.15	2.95 ^b^ ± 0.16
C20:4 n-6	32.35 ^a^ ± 0.98	34.91 ^a^ ± 1.85
C20:5 n-3	64.77 ^a^ ± 1.97	64.55 ^a^ ± 3.43
C22:4 n-6	3.30 ^a^ ± 0.10	3.50 ^a^ ± 0.19
C22:5 n-6	8.97 ^a^ ± 0.27	7.75 ^b^ ± 0.41
C22:5 n-3	18.35 ^a^ ± 0.56	22.09 ^b^ ± 1.17
C22:6 n-3	38.72 ^a^ ± 1.18	35.69 ^a^ ± 1.90
Total fatty acids	568.32 ^a^ ± 16.95	590.32 ^b^ ± 31.36
Σ SFA	118.56 ^a^ ± 1.63	121.53 ^b^ ± 7.22
Σ MUFA	244.10 ^a^ ± 8.31	269.51 ^b^ ± 6.01
Σ PUFA	205.66 ^a^ ± 7.00	199.29 ^a^ ± 11.84
Σ n-3 PUFA	121.84 ^a^ ± 4.15	122.33 ^a^ ± 7.27
Σ n-6 PUFA	83.82 ^a^ ± 2.85	76.96 ^a^ ± 4.57
n-3/n-6	1.46	1.60
DHA/EPA	0.60	0.55
IA	0.21	0.20
IT	0.18	0.18
PUFA/SFA	1.74	1.64
FLQ	18.20	16.98
mg EPA + DHA/100 g	681.98 ^a^ ± 321	554.33 ^b^ ± 5.43

^a,b^—Significant differences within each size group.

**Table 8 foods-12-03088-t008:** A summary of the average SFA, MUFA, and PUFA content {mg FA/gram of fat} and the values of the n-3/n-6 coefficient in different parts of crabs categorized by sex.

Index	Female			Male	
	Hepatopancreas	Meat	Gonads	Hepatopancreas	Meat
Σ SFA	251.19 ^aA^ ± 4.77	133.45 ^bA^ ± 2.65	120.04 ^c^ ± 4.42	218.25 ^aB^ ± 3.16	98.19 ^bB^ ± 2.09
Σ MUFA	342.37 ^aA^ ± 6.50	164.29 ^bA^ ± 3.27	256.80 ^c^ ± 12.16	263.16 ^aB^ ± 3.79	112.92 ^bB^ ± 2.46
Σ PUFA	235.75 ^aA^ ± 4.47	155.05 ^bA^ ± 3.06	202.47 ^c^ ± 9.42	243.64 ^aA^ ± 3.58	141.98 ^bA^ ± 3.05
Σ n-3 PUFA	87.56 ^aA^ ± 1.66	98.48 ^aA^ ± 1.93	122.08 ^b^ ± 5.71	76.36 ^aA^ ± 1.09	84.80 ^aA^ ± 1.85
Σ n-6 PUFA	148.19 ^aA^ ± 2.81	56.57 ^bA^ ± 1.13	80.39 ^c^ ± 3.71	167.28 ^aB^ ± 2.49	57.19 ^bA^ ± 1.20
n-3/n-6	1.70	0.57	0.66	2.25	0.68

^a–c^—Significant differences within each sex group. ^A,B^—Significant differences observed between sexes within the same body part.

## Data Availability

The data presented in this study are available on request from the corresponding author. The data are not publicly available due to [Patent procedure].

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
