# Peer review of "Nutritional Quality and Fatty Acids Composition of Invasive Chinese Mitten Crab from Odra Estuary (Baltic Basin)"

_foods, 2023, doi:10.3390/foods12163088_

Round 1

Reviewer 1 Report

Comments and Suggestions for Authors

This manuscript is about nutritional quality assessment of a new crab species for entrance to the European people food system. Some errors in the result report and discussion are seen in the manuscript. Also, some important health indexes of the fatty acids are not calculated. The manuscript suffers from lack of comparison by FAO/WHO recommendation for fatty acids daily intake.

The Detail of the reasons for major revision are listed below:

 Abstract: rewrite after corrections inside the manuscript.

Line 40: The reference number 10 not about 21st century and talk about 1995-1997 Chinese mitten crab in London, not Iran and Iraq. please check the reference.

Line 50: could you provide a new reference after 2020?

Line 63: " Regrettably, up until now....", start the sentence had strange sound, rewrite the sentence.

Line 84-85: Harvesting just in 10 days? for nutritional quality assessment it must harvest seasonally.

Line 87-88: how the crabs transported? in which temperature? with Ice or other method? what was the transportation time?

Line 95-97: exactly brief that in the meat separation and other processing methods how you preserved the claws, hepatopancreas and gonads? Temperature in the quality assessments of the fatty acids is very important.

Table 1: It is better to set "a" as better significant difference for the K factor of Male and "b" for the female. Also, write in the table caption that weight and carapace width don't show any significant difference in two species.

Table 3: use the "a" for the better data index and "b" and "c"... for the next good indexes. for example, in Meat yield.

Line 187: " as well as in the meat of males" is not true, because based on the table 4 data, the fat content in the male enhanced by increasing size.

Line 222-224: this is not clear. based on the table 6 significant differences observed on sex and size group. why you conclude:   but no significant differences were found based on size groups, contrary to what was observed in the hepatopancreas lipids.

Line 224-225: this is not true. n-3 PUFA also differ significantly in male by different size!!

Line 225-227: what is the meaning of this sentence? all data in female and male is below 100 mg Fa/g of fat. explain briefly.

Line 233-235: one of the gapes in this study is that you don't conduct statistical analysis between fatty acid contents in meat, hepatopancreas and the edible portions. specially in total MUFA, PUFA, n-3, n-6 and n3/n6. so, reader could not completely understand the differences between.

Line 236-241: this is not a true conclusion. for nutritional quality assessment must compare the n-3 PUFA of meat by the reference daily recommended standards. is this amount being higher than the daily intake or not?

Results:

one other gaps in this study are that you don't calculate the important health indexes in quality assessment of the fatty acids of the aquatic animals: Atherogenicity Index and Thrombogenicity Index. this is important that the fatty acid composition of the different part of this mitten crab is health for cardiovascular system or not!

Besides, other indexes are very important: PUFA/SFA, Unsaturation index, flesh Lipid quality, LA/ALA and Trans fatty acid (if measured).

your analysis not include C12:0 so you can’t calculate some of these indexes but others can be calculated based on your data and I recommend to do this for enhancing your manuscript quality.

Discussion

Line 266: reference 28 is old and you can bring newer one of 2022 or 2023.

Line 226-227: "...  there has been a shortage in the availability of aquatic resources"?!

it is better saying: high catch of aquatics and increase demand for aquatic foods culminate....

Line 269: Which species?

Line 269-274: It is better you start the discussion by your special results. these types of sentences could transfer to the introduction.

Line 286-288: this sentence is not related to the results and the findings must be discussed so can be eliminated.

Line 291-298: the goal and strait line of the discussion is not clear. Do you want to discuss about relation of your results to environmental conditions? or this article is about nutritional assessments of this crab fatty acids? I think it is better you change your vision to rewrite the discussion by the different lipid health indexes I mentioned in the result section comments.

Line 316-317: this is very important. you catch the crabs in just 12 days and the maturation of the gonads is in a special stage based on the annually growth period. so, the results can’t be generalized to the statistical community of the Chinese mitten crabs in this region. Also, this is confirmed in the line 321-323.

Line 389-399: it is not clear that do this new species have good n3/n6 ratio based on the FAO/WHO recommend for probable entrance to European people food or not?

Line 402-405: it is a theoretical calculation and could be true if you just prepared EPA+DHA for a human being from mitten crab. but it must be calculated in a food basket beside other food components same as fish, red meat, chicken, walnut, olive oil and so one. so, I think we could not conclude in this rigid way.

Line 408-411: this is not related to this study and could be eliminated.

Conclusion: must be rewrite based on the comments about health indexes of fatty acids.

References: must reoriented and corrected based on the journal style: for example, see the reference numbers 13, 26, …

Comments on the Quality of English Language

Moderate English language enhancing is need.

Reviewer 2 Report

Comments and Suggestions for Authors

This manuscript presents the composition and nutritional characteristics of the edible parts of a population of wild Chinese crabs from the Odra estuary in the Baltic basin. In addition, the study shows the differences in fat content and fatty acid composition in different parts of crabs according to sex and size.

Comments

Relevant articles from the last two years should be included in the introduction.

Authors should present in the article the size and weight distribution followed by male and female crabs.

It appears that about 70% of the crabs collected have a shell width between 60 and 75 mm. Why did the authors choose these categories and what about the middle 60-70mm group? How many crabs in each specific category (male, female, shell width 55-60 mm and 70-75 mm) were analyzed for fatty acids and fat?

In lines 251-253: The authors should try to explain the extremely higher content of palmitic acid in the gonads of the larger females compared to the smaller ones.

The significant digits of the standard deviation should not be more than two, and the mean value should have the same number of decimal places as their standard deviation. Please, correct the manuscript according to the comment above where necessary.

Comments on the Quality of English Language

Minor editing of the English language required

Round 2

Reviewer 1 Report

Comments and Suggestions for Authors

Dear Authors

You tried to correct the manuscript based on the comments. But the below comments are still alive:

Line 103-104: you mentioned “All samples of edible parts were preserved in ice (0oC) for further analysis”. It is not acceptable at all. For analysis of unsaturated fatty acids, samples must keep below -20 °C. You change the unsaturated fatty acid compositions before further analysis. Unfortunately, all of the next results are not acceptable.

Line 253-255 (233-235 old version): You write your opinion about your goal, but I still believe one of the gapes in this study is that you don't conduct statistical analysis between fatty acid contents in meat, hepatopancreas and the edible portions. specially in total MUFA, PUFA, n-3, n-6 and n3/n6.

-About compare the n-3 PUFA of meat by the reference daily recommended standards, you write an answer “We consider this to be a true conclusion. Taking into account the FAO/WHO recommended daily intake of EPA and DHA acids of 500mg (containing in the range of 0.250 - 2g), the amount provided by crab muscles is about 10 times lower. This is also evidenced by the indicator of the sum of mg EPA and DHA added to the tables per 100g of the product. According to the Food and Agriculture Organization of the United Nations (UN FAO), the recommended amount is 0.250–2 g/day.” But this is not seen inside the body of the revised manuscript.

- in the line number 47, you say “A total of 17 fatty acids were identified and quantified…” so your standard injected to the instrument just had these 17 fatty acids. You don’t have opportunity to calculate C12:0 fatty acid. So how you calculated IA? You say in answer that set zero for this fatty acid inside the formula, but this is not correct.

- About IA and IT, in the discussion section you just set comparison by other aquatics, but this is not significant that these fatty acid compositions are dangerous for cardiovascular system or not.

- Line 503: Rincon-Cervera at all?!!

Reviewer 2 Report

Comments and Suggestions for Authors

The paper has been improved significantly.

Author Response

Dear Reviewer,

We would like to thank you for the review and accepted our work.

Sincerely,

Grzegorz Tokarczyk